# The Effectiveness of a Novel Air-Barrier Device for Aerosol Reduction in a Dental Environment: Computational Fluid Dynamics Simulation

**DOI:** 10.3390/bioengineering10080947

**Published:** 2023-08-08

**Authors:** Xiaoting Ma, Won-Hyeon Kim, Jong-Ho Lee, Dong-Wook Han, Sung-Ho Lee, Jisung Kim, Dajung Lee, Bongju Kim, Dong-Myeong Shin

**Affiliations:** 1Department of Mechanical Engineering, The University of Hong Kong, Pokfulam 999077, Hong Kong; xtma_2020@connect.hku.hk; 2Dental Life Science Research Institute, Seoul National University Dental Hospital, Seoul 03080, Republic of Korea; wonhyun79@gmail.com (W.-H.K.); shlee79@snu.ac.kr (S.-H.L.); jisung920415@naver.com (J.K.); dajueng90@gmail.com (D.L.); 3Daan Korea Corporation, Seoul 06252, Republic of Korea; daankorea@naver.com; 4Department of Cogno-Mechatronics Engineering, College of Nanoscience and Nanotechnology, Pusan National University, Busan 46241, Republic of Korea; nanohan@pusan.ac.kr

**Keywords:** air-barrier, aerosol, computational fluid dynamics (CFD), contaminated particles, dental environment

## Abstract

The use of equipment such as dental handpieces and ultrasonic tips in the dental environment has potentially heightened the generation and spread of aerosols, which are dispersant particles contaminated by etiological factors. Although numerous types of personal protective equipment have been used to lower contact with contaminants, they generally do not exhibit excellent removal rates and user-friendliness in tandem. To solve this problem, we developed a prototype of an air-barrier device that forms an air curtain as well as performs suction and evaluated the effect of this newly developed device through a simulation study and experiments. The air-barrier device derived the improved design for reducing bioaerosols through the simulation results. The experiments also demonstrated that air-barrier devices are effective in reducing bioaerosols generated at a distance in a dental environment. In conclusion, this study demonstrates that air-barrier devices in dental environments can play an effective role in reducing contaminating particles.

## 1. Introduction

The global demand for clean air in confined places is increasing rapidly with the spread of hazardous microorganisms. Dental procedures involving the use of high-speed instruments such as tooth cleaning, scaling, and drilling, generate aerosols consisting of microorganisms [1,2,3]. Because microbial species in dental treatment include pathogenic bacteria, viruses, and fungi [4], frontline dental personnel are exposed to serious risks of cross-transmission. Studies have shown that microorganisms with diameters less than 5 μm can be airborne for a few hours over a long distance (a few meters) [4,5]. According to a study, chest areas have the highest microbial contaminant concentration, and the contaminants can spread up to a radius of 800 mm from the patient’s head [4], hence, prompting researchers to develop technologies to quickly and effectively ventilate the air around the patient and medical staff.

To date, numerous technologies, including high-volume evacuators (HVEs) [2,6,7,8], rubber dams [9,10], face masks [11,12], and preprocedural mouth rinses [2,13], have been used to reduce the formation and spread of microbial particulates in dental practices. The high volume evacuator has been reported to reduce airborne microbial contamination by 90% [2,8,9]; however, it requires extra manpower (due to difficulty in use) and results in a higher risk of cross-infection. Moreover, the HVE must be located close to the surgical site (within 10 mm), causing additional inconvenience and discomfort. Even though the rubber dam helps to minimize the microorganisms released from the mouth into the atmosphere [9,10], it hinders medical treatment owing to its inherent screening mechanism. Although several types of personal protective equipment such as face masks, gloves, and goggles are designed to mitigate medical place hazards, the risk of respiratory infection transmission remains. In addition, through previous studies, high levels of infection risk have been reported within 50 cm of the patient’s head during dental procedures, underscoring the importance of infection control measures for dentists, nurses, and equipment surfaces within the area close to the patient’s head [4,14,15]. The HVE has low microbial interception capabilities in the area proximal to the patient’s head, necessitating the development of a new air-barrier that can be attached to a dental-light device or placed near dental chairs. Hence, new technologies that exhibit fast removal rates, high removal capabilities, and ease of use are needed. An air curtain is a well-developed technology that is used for the elimination of mass or heat transfer through the opening without being a physical barrier [16,17], applied to an invisible firewall [18], entrance [17], and smoke blocker [19,20]; however, this technology has yet to be applied to personal protective equipment.

To solve these problems, we developed an air-barrier device that can be assembled into a dental-light device. A computational fluid dynamics (CFD) simulation was performed to derive the improved design of an air-barrier prototype that can effectively remove contaminants, such as bioaerosols. In addition, an experimental environment was constructed to verify the droplet shielding effect of the improved prototype; the performance was evaluated before and after the air barrier was activated, and a comparative analysis was performed. Therefore, the purpose of our study was to derive a better design of an air-barrier device and identify the droplet-blocking effect by evaluating the performance of the prototype.

## 2. Materials and Methods

### 2.1. The CFD Simulation for the Better Design of Air-Barrier Devices

CFD [21,22] was used to simulate the airflow behaviors in the ventilation systems. A standard k-ε turbulence model in COMSOL Multiphysics^®^ was adopted to simulate the average turbulent airflow by solving the Reynolds-averaged Navier-Stokes (RANS) equation. The k-ε model is well suited for flows with planar shear layers and recirculation zones. It is the most common and reliable turbulence model for various applications in industry and environment. In the study, we mainly investigated the recirculating flows that functioned as air barriers and removed the bioaerosols in a clinic environment. Furthermore, the droplet blocking effect and bacterial barrier effectiveness experiments also validated the rationality of using the k-ε model in the simulation. The fluid (air) was set to be compressible (the highest pressure reached 1.44 atm in this study) and the wall treatment was with wall functions. The Reynolds numbers of the chamber of the device was estimated to be over 4900, which indicated the formation of turbulent flow in the simulations. Because the device was supposed to be a metal material, the wall condition was set to no slip. For the inlet, a fully developed flow condition was selected, and the input variable was the average velocity; for the outlet, a velocity condition was selected, and the input variable was the normal outflow velocity. A physics-controlled mesh with a fine element size was applied, and a wall resolution in the range of 12–24 m^2^ was well guaranteed. For the meshing, the maximum element size was 0.0105 m and the minimum element size was 0.0005 m, and the total number of elements was 632,455 to 1,556,138 depending on the distance between the nozzle and patient’s mouth. Moreover, the airflow simulation was conducted as a stationary study and particle tracing simulation was a time-dependent study. The gird resolution in Grid 1D, Grid 2D, or Grid 3D was 1000, 100, and 30, respectively. The convergence and mesh independence study were shown in Appendix A.

The tracing of microorganisms was realized by the particle trajectory module, which provides an effective way to investigate the particle transmission behaviors in airflow, assuming that the impact of the particles on the airflow field is negligible. The motion of the particles was calculated using Newton’s second law:(1)$$m\frac{d^{2}x}{dt^{2}}=F\left(t,x,\frac{dx}{dt}\right)$$
where *x* is the particle position, m is the particle mass, and F is the total force applied to the particle. The particles were released at a rate of 50 per 0.1 s in the first second, with an initial upward velocity of 0.2 m/s. The total number of particles released was 550 and the density of the particles was 1 g/cm^3^. Although the size of the particulates generated in dental procedures is controversial, ranging from several microns to hundreds of microns, we selected particles with diameters ranging from 1 μm to 5 μm in this study. The wall condition was set as “freeze” since the particles generated in the dental procedure were generally droplets or droplet nuclei. For the drag force of the simulation, we utilized Stokes law, assuming that:(2)$$F=\pi r^{2}\rho \left|u-u_{p}\right|(u-u_{p}){\left[1.84Re_{P}^{-0.31}+0.293Re_{P}^{0.06}\right]}^{3.45}$$
(3)$${Re}_{p}=\frac{\left|u-u_{p}\right|2r\rho }{\mu }$$
where F denotes the drag force [23,24], Re*_p_* the Reynolds number, u the velocity of the fluid, u*_p_* the velocity of the particle, r the particle radius, ρ the fluid density, and μ the dynamic viscosity of the fluid.

### 2.2. Experiment Set-Up of Droplet Blocking Effect for Evaluation of Air-Barrier Device

Modeling and drawings were constructed using a 3D CAD program (Solidworks 2016, Dassault Systemes SolidWorks Corp., Waltham, MA, USA). A prototype was set such that the air exited in the 25.3° direction, and the inner space was designed with a diameter and height of 70 and 50 mm, respectively (Figure 1a). In addition, six small hose connectors were constructed to receive air from the compressor (Figure 1b). Based on the drawing data, a prototype was manufactured using a 3D printer (Object 260 Dental, Stratasys Ltd., Eden Prairie, MN, USA) as shown in Figure 1c,d. 

Similar to the distance between the dental light and patient in the dental treatment environment, the air barrier was fixed to the upper end of the experimental table, and the distance from the oral cavity phantom model was set at 500 mm (Figure 2A–C). To evaluate the droplet-blocking effect, the droplet particles detected from the detection paper on the side of the experimental table were photographed using the same conditions and camera equipment. In addition, the number of particles was analyzed using ImageJ software (developed by Wayne Rasband; http://rsb.info.nih.gov/ij/index.html (accessed on 31 October 2022), provided in the public domain by the National Institutes of Health, Bethesda, MD, USA). High-speed camera (HS7-H, Fastec Imaging Corp., San Diego, CA, USA) images were taken at 1000 frames per second [25] for 2 min to confirm the droplet-blocking effect of the air-barrier device. The time during which the air-barrier device was operated was divided equally by 1 min.

### 2.3. Experiment Set-Up to Evaluate Bacterial Barrier Effectiveness of Air-Barrier Device

The experimental environment was constructed in the same way as the droplet blocking effect condition (Figure 2), and two cultured media were attached to each of the four sides to evaluate the barrier effect of the bacteria (Figure 2D). The standard strain used in this study is *Streptococcus mutans* (*S. mutans*, KCOM 1128), and the strain was isolated from the gingival bacterial membrane of Korean people and purchased from the Korean Collection for Oral Microbiology (Gwangu, Korea). The *S. mutans* was incubated in a bacterial incubator at 37 °C, 5% CO_2_ for 24 h using a brain heart infusion (BHI, Becton, Dickinson and Company, Sparks, MD, USA) liquid medium and agar medium [26,27]. The *S. mutans* was diluted 1:1000 in PBS and applied to a portable handpiece to analyze the effectiveness of the air curtain in blocking droplets by comparing the amount of the *S. mutans* detected on agar media attached to the left, right, and front of the experimental table before and after the air curtain was operated [28]. In addition, this study confirmed that the air blown from the air-barrier device in the box space with a distance of 500 mm horizontally and vertically is directed to the dental drape covered by the oral cavity phantom model without touching the box on the side. Therefore, it was confirmed that the box space does not affect the air flow from the air-barrier device in the experimental environment.

### 2.4. Statistical Analysis

All data were processed using the SigmaPlot 14 statistical program (Systat Software Inc., San Jose, CA, USA) and expressed as mean ± standard deviation (SD). In addition, the data were evaluated using the Mann–Whitney rank-sum test. The results were considered significant if the *p*-value was less than 0.05.

## 3. Results and Discussion

### 3.1. The Results of CFD Simulation of Air-Barrier Devices

Figure 3 illustrates the air-barrier device component and basic concepts of this study. An air-barrier device featuring an air-jet injector and ventilation hole was built into the overhead dental light, as shown in Figure 3A. The air jet ejected from the overhead dental light creates an invisible air barrier in the vicinity of the patient’s mouth. The air barrier divides space into two different zones, including the patient and medical staff zones, preventing the migration of aerosols across these zones (Figure 3B). Aerosol contaminants from the patient were evacuated through the ventilation hole at the center of the dental light. In this study, the overhead dental light implementing the air-barrier device was configured with a cylindrical structure (diameter and height of 80 mm and 115 mm, respectively, with the chamber’s height at 55 mm, four inlets of diameter 2 mm, one ventilation hole of diameter 6 mm, and the opening of an aerojet injector at 2.5 mm (See the details in Appendix A) with a four-fold symmetry in which the air chamber was integrated with a fresh-air inlet and air-jet injector, as depicted in Figure 3C. The ventilation hole was connected to an external pump to collect contaminants from patients. The boundary condition of the ventilation hole is normal outflow velocity varying in the range of 0–20 m/s for different individual simulations. A facial model, in which a rigid block and an opening are modeled as the skin and mouth, respectively, was located underneath the air-barrier device at a specific distance. A large volume, at least twice the device size, was set as an open boundary at atmospheric pressure. A commercial CFD with the k-ε turbulence model was used to predict the flows in the patient zone, assuming that the fluid is compressible. The wall condition was set to be non-slip because the device was assumed to be made of metallic materials. For the fresh-air inlet, a fully developed flow condition with a variable average inlet velocity was chosen.

Figure 4A exhibits the representative velocity distribution of the air barrier. The facial model was placed 100 mm away from the air-barrier device, in which the air jet was injected at a velocity of 5.8 m/s and an angle of 75° with respect to the bottom surface of the system, and no ventilation velocity was applied. In addition, a strong air stream was generated along the direction of the injection nozzle, indicating the successful formation of an air barrier. Some of the important features are that downward flow occurs in the proximity of the air barrier, whereas upward flow is found in the area away from the air barrier, leading to re-circulation in the patient zone. The velocity and width of the air barrier were recorded along the central line of the stream (Figure 4B). From the analytical models of the air barrier [16,19,29,30], we identified three distinguishable regions: (1) potential core, (2) transition, and (3) developed zones. In the potential core zone, the centreline velocity is almost constant and close to the initial speed in the nozzle; in the transition zone, the velocity starts to decay and the jet expands, generally starting at approximately 5 o (o—characteristic dimension of nozzle) from the nozzle; in developed zone, velocity decay remains constant, generally starting at approximately 20 o from the nozzle.
(4)$$V_{c}=\frac{1}{2}\left[1+erf\left(C_{1}\frac{0.5o}{x}\right)\right]V_{i}$$
where *x* is the distance from the nozzle, *o* is the nozzle size, and *C*_1_ is a constant (*C*_1_ = 3.65, for this study). At a distance of approximately 37 mm from the nozzle, the velocity decay is proportional to the inverse square root of the distance from the nozzle in the developed zone.
(5)$$\left(V_{c}=C_{2}V_{i}{\left(\frac{x}{o}-C_{3}\right)}^{-1/2}\right)$$
where *V_i_* is the initial velocity, and *C*_2_ and *C*_3_ are constants. The values for *C*_2_ and *C*_3_ are 1.86 and 4.21, respectively, which are consistent with a previous report [31]. The average dispersion of the air jet gradually increased with increasing distance from the nozzle, exhibiting the highest width of 20.6 mm (eight-folds greater than the nozzle width) at a distance of 100 mm from the nozzle. The velocity distributions of the air jet could be tuned by varying parameters such as the fresh-air inlet velocity as well as the air jet injection angle when no ventilation was applied. Increasing the fresh-air inlet velocity from 50 to 90 m/s increased the average velocity of the air jet (Appendix A). A high injection angle (75°) resulted in the lowest average velocity of the air jet, with an increase in average velocity for a decrease in injection angle (Figure 4C). This resulted in an enhanced velocity up to approximately 280% for a 64.7° injector compared to the velocity for a 75° injector at a distance of 100 mm from the nozzle. It should be noted that the air-jet injection velocity was modulated by altering the injection angle, even at a constant velocity of the air inlet, benefiting the device in terms of energy savings for flow generation. Consequently, we successfully demonstrated a tunable air barrier in a given region by controlling each parameter that affects the kinetics of the air jet.

By operating the ventilation system, we observed that the velocity distributions of the air jet did not vary with the ventilation velocity up to 10 m/s (Appendix A); however, the streamlines experienced a significant change when strong ventilation of 20 m/s was applied (Appendix A). At a low ventilation velocity, an upward flow was generated in the patient zone owing to recirculation, presumably enabling the driving of buoyancy drag forces to contaminants in the vicinity of the patient’s mouth. However, the rapid ventilation-induced pressure drops at the top of the patient zone so that the air barrier tends to distort towards the ventilation hole and diffuse into the patient zone, resulting in an upward flow (Appendix A). Figure 4D shows the average velocity as a function of the vertical distance from the device at different injection angles. For all injection angles, a downward flow towards the patient’s face was observed at strong ventilation of 18 and 20 m/s. Notably, a downward flow was turned into an upward flow near the device even under strong ventilation by modulating the injection angle, implying that the combination of injection and ventilation velocities needs to be adjusted for different working distances.

The transmission behaviors of contaminants were simulated by outpouring spherical particles into the patient zone from the patient’s mouth. In one second, 550 particles with an initial upward velocity of 0.2 m/s were discharged at a rate of 50 particles per 0.1 s. Although the size of particulates generated in dental procedures remains controversial [14,32,33,34], we selected particles with diameters in the range of 1 μm to 5 μm in this study, as the removal efficiency of particles with diameters less than 4 μm has dramatically decreased in conventional filtration technology [35]. Figure 5A illustrates the modeled processes that affect the calculation of the removal efficiency of the released particles. All microparticles were frozen once they reached any boundary or device. As we investigated the effect of the air-barrier device on the spreading of contaminants into the medical staff zone, only particles drifting in the air were considered for analysis; therefore, the removal efficiency of particles was determined by
(6)$$\eta_{removal}=\frac{N_{vent}}{N_{vent}+N_{open\;boundary}}$$
where *η_removal_* is the vented particles (%) and N*_vent_* and N*_open_* _*boundary*_ correspond to the number of particles frozen at the ventilation outlet and the particles injected from the mouth, respectively.

The rate of contaminant removal from a region is a key property that determines the processing time, and the rate of microparticle removal from the patient zone was evaluated using a device with a constant injection angle of 75° and ventilation velocity of 5 m/s. As shown in Figure 5B and Appendix A, the devices with varied air-barrier injection velocities (from 3.1 m/s to 5.1 m/s) achieved their maximum removal efficiency of 1 μm particles within 3 s. The time to reach maximum efficiency (~99%) was shortened with injection velocity, resulting in a fast removal time of 2.5 s when the injection velocity was 5.1 m/s. In addition, the fast air-barrier injection led to a higher maximum removal efficiency, with a maximum efficiency of up to 99.2 % at 5.1 m/s injection velocity. This result was expected owing to the fast drift flow near the patient’s mouth. The concept of the contaminant removal effectiveness (CRE, *ε*) index, given by Equation (7) [36,37], was employed to quantify the air quality in the patient zone, and the results are presented in Appendix A.
(7)$$\varepsilon =\frac{C_{e}-C_{s}}{C-C_{s}}\approx \frac{C_{e}}{C}$$
where *C_e_* and *C_s_* are the particle concentrations in the ventilation outlet and the supply air, respectively. C represents the average particle concentration at the open boundary. Overall, the CRE indices exhibit much greater values than 1, which indicates that the discharged contaminant sources are not recirculated in the patient zone. The rate of contaminant removal from a region was further improved by inclining the injection nozzle from 75° to 64.7° and extracting the particles at the ventilation outlet with velocities varying from 2 m/s to 10 m/s (Figure 5C and Appendix A). The fastest particle removal of 2 s was obtained at an injection angle of 64.7° and a ventilation velocity above 4 m/s. In this study, it is worth noting that the CRE index increased 10 times at the best-performing condition, even though the time to reach the saturated removal efficiency was only 0.5 s shorter (Appendix A), indicating that the low injection angle and fast ventilation velocity benefit diluting contamination in the patient zone. Furthermore, maximum removal efficiency diminished by 9 points in differing particle sizes from 3 μm to 1 μm (Figure 5D and Appendix A); however, it still showed that 9 out of 10 particles could be effectively removed from the patient zone. This trait is particularly advantageous, given that the removal efficiency has been shown to decrease dramatically when the contaminant particles are small in conventional technology [35].

Furthermore, we simulated the performance of an air-barrier device at a reasonable distance (500 mm) from the patient’s mouth to secure sufficient working space for the practitioner in a real dental operation. The removal efficiency rates obtained at an injection angle of 64.7°, ventilation velocity of 2 m/s, and different injection velocities are shown in Figure 6A. A well-defined removal curve was observed for all the injection velocities. The removal saturation time of 26.8 s was achieved at the injection velocity of 12 m/s, and this value decreased to 22.5, 20.9, and 18.8 s at the velocities of 13, 15, and 16 m/s, respectively. A long-distance likely gives rise to a long pathway to reach the ventilation hole, thereby increasing the removal time compared to the short distance (100 mm). However, their CRE indices achieved infinity, which means saturated removal efficiency was 100%, demonstrating that the device effectively prevents the recirculation of contaminants in the patient zone (Appendix A). For comparison, we also simulated a dental suction device placed 10 mm above the patient’s mouth and evaluated its removal efficiency rate (Figure 6B). At the same ventilation velocity of 2 m/s, the removal time was substantially longer (~12 s) than that of the device, which was located 10 mm above the patient. Furthermore, the actual removal time, which corresponds to the time after the turn-on of removal, even at a faster ventilation velocity of 3 m/s, is two-fold longer than that of the devices installed 500 mm above the patient. As the patient continuously discharges contaminants during dental treatment, this result preliminarily suggests that the device is beneficial for removing contaminants not only quickly but also effectively.

### 3.2. The Results for the Droplet Blocking Effect of the Air-Barrier Device

In this study, the prototype device was designed based on the factor of improved air-barrier design derived through CFD results. In recent years, studies have been conducted for the analysis of droplet particles. In addition, some studies have analyzed the exposure of dental personnel such as dentists and dental hygienists [4,25,38]. Another study used an oral phantom model and dental handpiece to evaluate the particles and droplets injected into the oral cavity of the phantom model. These studies placed the subject or measurement target near the dental chairside and evaluated droplet infection. In another study, an oral phantom model and dental handpiece were used to evaluate the particles and droplets ejected from the oral cavity of the phantom model [9,39]. In order to confirm whether the air-barrier prototype was able to block droplets, we constructed an experimental table in a form similar to the dental environment (Figure 2A). Additionally, the experiment was performed using an oral cavity phantom model and dental handpiece, as described in previous studies (Figure 2B). Previous studies have reported that droplets are ejected first in the front and top of the oral cavity and then spread to the sides and back in sequence [40]. Therefore, in this study, detection paper was pasted on four surfaces of the experimental table for the detection of droplet particles (Figure 2A). Moreover, fluorescent material was used to analyze the particles ejected by an oral phantom model with a dental handpiece operating at 300,000 rpm (Figure 2C). The evaluation of the air-barrier prototype was classified into two groups: (1) before and (2) after the air-barrier operation, and the results measured in three repeated experiments were calculated and compared as an average value. In a previous study, droplet particles were measured and analyzed [41,42]. Therefore, in this study, the number of particles was collected and comparatively analyzed to evaluate the air-barrier effect.

The dental handpiece was driven into the oral cavity of the phantom model to eject droplets, such as bioaerosols, while detection papers were pasted on four surfaces of the experimental table for the detection of droplets. Table 1 shows the number of droplets on four surfaces of the experimental table before and after the operation of the air-barrier device. Most droplet particles were detected in the front area of the oral cavity phantom model. By contrast, a much smaller amount was found both for the left and right sides, and even no droplets were detected in the rear area.

The average value was 359.25 ± 804.45 in the case where the air-barrier device was not operated (Table 2). However, in cases where an air-barrier device was used, the average number was only 0.5 ± 1.0 (Table 2). According to this result, the ratio between droplets generated after the air-barrier operation and those generated before the operation of the air barrier was 0.14%, which means the effect of the blocking effect of the prototype for droplets was more than 99%. A statistically significant (*p* = 0.028) lower value was observed after the operation than before the operation of the air-barrier device (Table 2).

Previous studies have reported that droplets were generated in forward and upward directions when the patient was lying down [38,40,43]. Similar to previous studies, the study also confirmed that the droplets generated through the oral cavity phantom model ejected forward and upward (Figure 7A–D). Furthermore, we confirmed that the droplets captured by the high-speed camera were ejected more than 30 cm from the oral phantom model before the air-barrier device was operated (Figure 7A,C). However, the droplets remained only around the oral phantom model after the air-barrier device was operated (Figure 7B,D). As a result, the air-barrier device was considered to be effective for blocking droplets. A previous study reported that a large number of droplets were initially dispersed, spread forward, sideways, and backward over time, and a large number of dispersed droplets were detected [40]. Thus, the initial control of droplet particles is important for infection prevention.

### 3.3. The Results for Bacterial Barrier Effectiveness of the Air-Barrier Device

To evaluate the effects of external factors, five experimental conditions were considered. Observations of the five external factors showed that no microbials were observed in any group (Figure 8) [28,44,45].

In the bacteria detection results, droplets containing *S. mutans* were observed on the BHI agar plates before the air-barrier device was used. The highest number of *S. mutans* was observed on the agar medium attached to the front, and *S. mutans* was also observed on the agar medium on the left and right sides (Figure 9).

After using the air-barrier device, some *S. mutans* were observed on the front, left, and right BHI agar plates, but most *S. mutans* were blocked after using the air barrier device compared to before using the air-barrier device. The results of *S. mutans* counting eight BHI agar plates per group showed that the bacterial count value exposed to *S. mutans* droplets before using the air-barrier device was 128.25 ± 221.91 CFU/cm^2^, and the *S. mutans* count value on agar plates after using the air-barrier device was 0.5 ± 0.53 CFU/cm^2^ (Table 3).

The analysis of the total area of *S. mutans* cultured from droplets before and after the air-barrier device was applied showing that the percentage of total area occupied by the bacteria relative to the diameter of the agar plates exposed to *S. mutans* droplets without the air-barrier device averaged 21.56 ± 15.79%, while the percentage of total area occupied by the bacteria relative to the diameter of the agar plates with the air-barrier device blocked droplets averaged 0.001 ± 0.001%. Based on the combined results of the droplet blocking experiments with and without the air-barrier device, *S. mutans* was observed to be most exposed on the front, with exposure on the left, right, and back. When the air-barrier device was applied, some very small amounts of *S. mutans* were exposed, but the droplet blocking rate was 99.61% according to the bacteria count, and the droplet blocking rate was 99.99% according to the bacteria detection area. Previous studies have shown that spatter is a solid particle with a particle size of about 100 µm or more and can contaminate the dental work area, skin, eyes, and clothing, and aerosols are smaller than about 50 µm and can contaminate the entire surgical environment through airflow [1,46,47]. Ionescu et al. reported that aerosols are carriers of infectious agents that contaminate all exposed surfaces, including medical staff, patients, and the surgical environment, and in the distribution of surface contamination in dental operating rooms, contamination is observed on all parts of the dental unit chair [28]. We observed that the bacteria in the droplets applied to the phantom model could not pass through the air-barrier device operated in a limited area, confirming that the patient, medical staff, and dental unit chair can be controlled from the source of contamination.

In this study, an improved design was derived through CFD simulation, and a prototype model was manufactured to identify whether bacteria could be blocked. Based on the CFD simulation and experimental results, the effectiveness of the air-barrier device was confirmed. However, the material used in the prototype was only a biocompatible liquid polymer material, which is a 3D printing material. Since the surface characteristics of the prototype model may change depending on the 3D printing material, it is necessary that future research should consider various materials applicable to the air-barrier device to verify the effectiveness of bioaerosol blocking due to surface changes.

## 4. Conclusions

This study has reported on an air-barrier device that exhibits effective performance as personal protective equipment in the dental clinic, with a rapid particle removal rate, high removal efficiency, and exceptional contaminant removal effectiveness. By adjusting several parameters, including injection velocity, injection angle, ventilation velocity, and distance, we derived an improved design of the air-barrier device with better removal rate of the droplet compared with a conventional suction device. The study demonstrated that the air-barrier device can provide an excellent platform for the development of high-performance protective equipment by combining it with ventilation apparatus. The air-barrier prototype presented in the study exhibits remarkable removal rates of microparticles and high removal efficiency, together with extreme contaminant removal effectiveness, suggesting that the device can serve as efficient protective equipment for dental personnel.

## 5. Patents

Seoul National University Dental Hospital has applied for a patent (KR application No. 10-2018-0034366 (WIPO application No. PCT/KR201/001800) and 10-2019-0082272) on some of the technology discussed here, on which J.-H.L., B.K., M.-Y.K. and W.-H.K. are listed as coinventors.

## Figures and Tables

**Figure 1 bioengineering-10-00947-f001:**
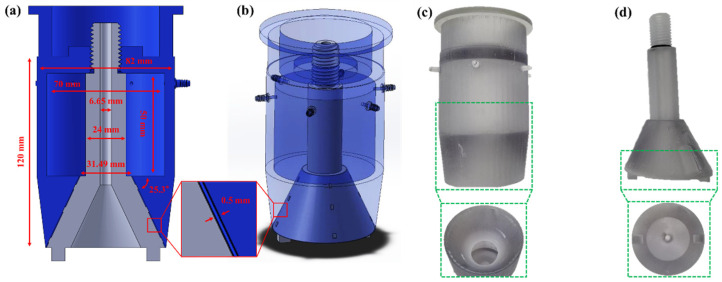
3D modeling and prototype of the air-barrier device. (**a**) cross-sectional view and dimensional information of the air-barrier 3D model. (**b**) three-dimensional and perspective view of the 3D air-barrier model. Shape and internal structure of (**c**) upper and (**d**) lower parts for the air-barrier prototype.

**Figure 2 bioengineering-10-00947-f002:**
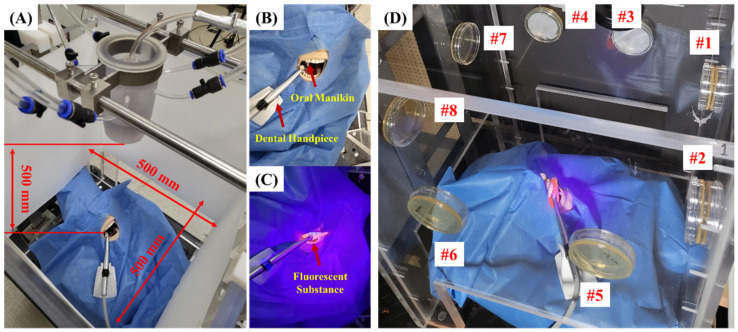
Schematic illustration of the experiment for the bioaerosol shielding effect of air-barrier device. (**A**) Schematic illustration of the experiment for evaluating the bioaerosol blocking effect of the air-barrier device. Application of the fluorescent substance and UV light to analyze bioaerosol particles ejected from an oral phantom model. (**B**) States for not applying UV light. (**C**) States for UV light irradiation. (**D**) Location of media attachment for detection of bacterial strains; front: (#1–2), left: (#3–4), right: (#5–6), back: (#7–8).

**Figure 3 bioengineering-10-00947-f003:**
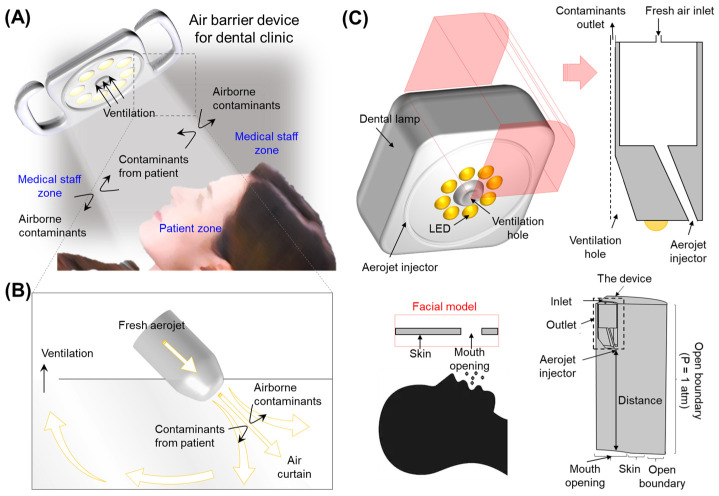
Air-barrier device. (**A**) Schematic illustration of a dental light equipped with the air-barrier device. The air-jet injector in the air-barrier device generates an invisible air barrier, sectioning the space into the patient zone and medical staff zone. (**B**) Aerojet acts as an air barrier to prevent the migration of aerosols across the zone. (**C**) Schematics of the air-barrier device in which the aerojet injector and ventilation hole are included.

**Figure 4 bioengineering-10-00947-f004:**
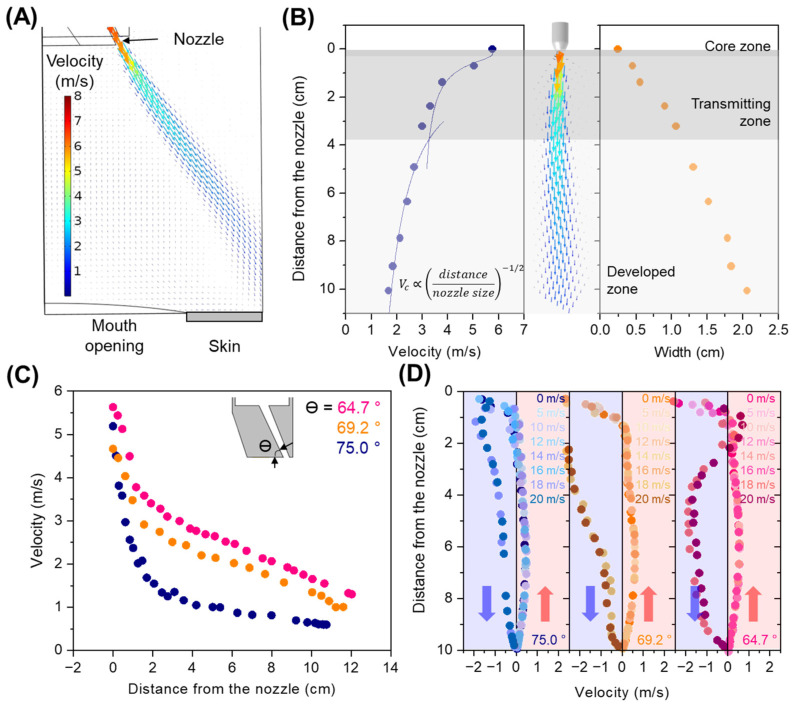
Airflow in the operation of the air-barrier device in COMSOL simulation (with no ventilation). (**A**) Streamlining of the air barrier. (**B**) Velocity and width of the air barrier as a function of distance from the nozzle. (**C**) Velocity profiles of the air barrier at different injection angles. (**D**) Velocity profile along the vertical direction in the patient zone (the pink arrows indicate the airflow is in an upward direction, while the purple arrows indicate the airflow is in a downward direction).

**Figure 5 bioengineering-10-00947-f005:**
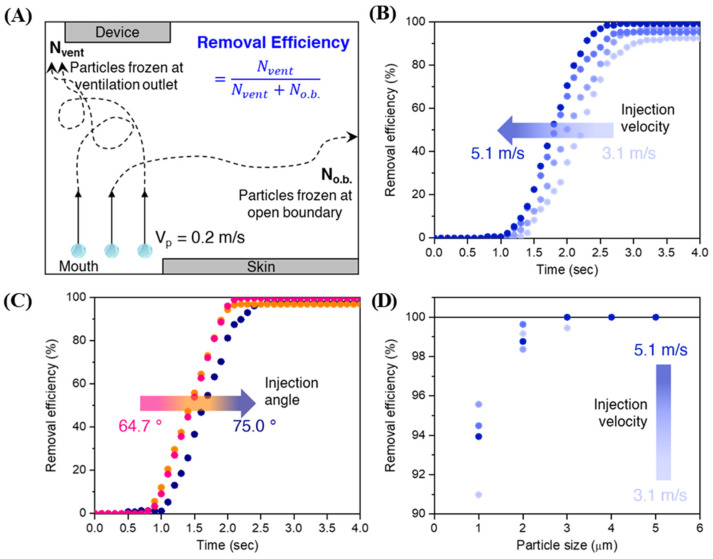
Microparticles evacuation. (**A**) Schematic illustration of calculation for vented particles (%). (**B**) Rates of vented particles (%) at varied injection velocities. (**C**) The removal efficiency as a function of time at differing injection angles. (**D**) The removal efficiency of microparticles in the range of 1 μm to 5 μm at different injection velocities.

**Figure 6 bioengineering-10-00947-f006:**
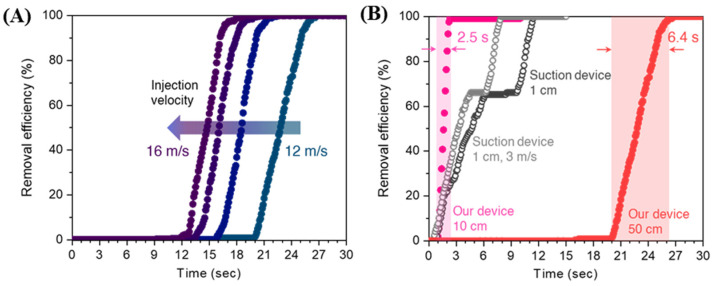
Air-barrier device prototype tests. (**A**) Vented particles (%) at a reliable distance of 500 mm. (**B**) Rates of removal efficiency of the prototypes and suction devices.

**Figure 7 bioengineering-10-00947-f007:**
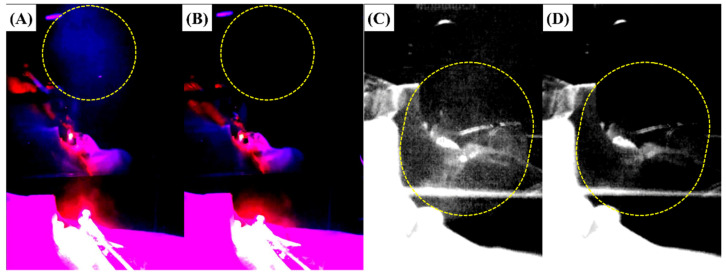
Observation of bioaerosol trajectory using high-speed camera when the air-barrier device was off (**A**,**C**) and on (**B**,**D**). (**A**,**C**) Lots of bioaerosol ejected from the oral cavity and diffused to the outside when the air-barrier device did not operate during the operation of the dental handpiece. A UV light and fluorescent substance was used to emphasize the bioaerosol captured by a high-speed camera. (**B**,**D**) Almost no bioaerosol was observed during the operation of the dental handpiece when the air-barrier device was activated.

**Figure 8 bioengineering-10-00947-f008:**
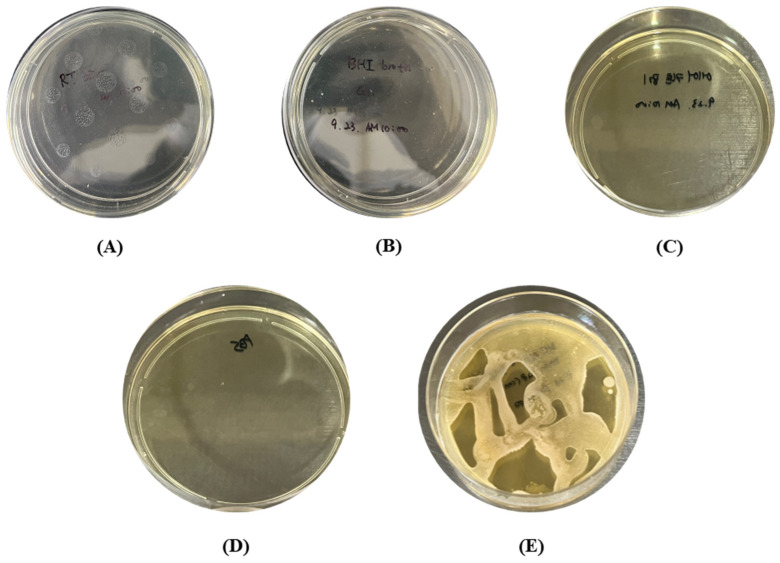
The results for the effects of five external factors. (**A**) exposure to air in the experimental space, (**B**) liquid BHI medium, (**C**) exposure to air from the air-barrier device, (**D**) contamination in the handpiece hose, (**E**) presence of the *S. mutans* in the medium.

**Figure 9 bioengineering-10-00947-f009:**
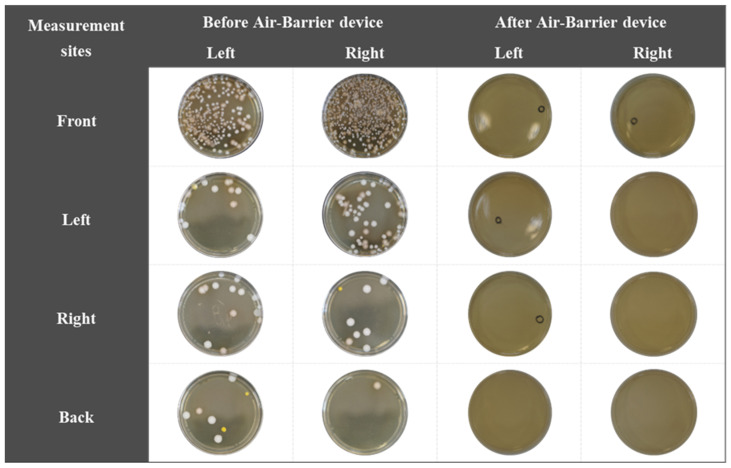
Detection results of bacteria in two media attached to each of the four sides of the experiment table before and after using the air-barrier device.

**Table 1 bioengineering-10-00947-t001:** The number of droplet particles detected on the four surfaces of the experimental table before and after the operation of the air-barrier prototype.

Group	Repetition	Side	Number of Particles
Before air-barrier	1	Front	305
		Left	12
		Right	6
		Back	0
	2	Front	1394
		Left	15
		Right	4
		Back	0
	3	Front	2575
		Left	0
		Right	0
		Back	0
After air-barrier	1	Front	3
		Left	0
		Right	0
		Back	0
	2	Front	0
		Left	1
		Right	2
		Back	0
	3	Front	0
		Left	0
		Right	0
		Back	0

**Table 2 bioengineering-10-00947-t002:** The results of droplet particle count before and after the air-barrier prototype.

Group	N	25%	75%	Mean (Median)
Before air-barrier	12	0.00	232.50	359.25 (5.00)
After air-barrier	12	0.00	0.75	0.50 (0.00)
*p*-value between two groups				0.028 *

* Statistically significant (*p* < 0.05).

**Table 3 bioengineering-10-00947-t003:** The results of droplet particle count for bacteria before and after the air-barrier prototype.

Measurement Sites		Before Air-Barrier Device (CFU/cm^2^)	After Air-Barrier Device (CFU/cm^2^)
Front	#1-Left	364	1
	#2-Right	585	1
Left	#3-Left	14	1
	#4-Right	32	0
Right	#5-Left	15	1
	#6-Right	8	0
Back	#7-Left	7	0
	#8-Right	1	0

## Data Availability

The datasets generated during and/or analyzed during the current study are available from the corresponding author on reasonable request.

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
