# Peer review of "The Effectiveness of a Novel Air-Barrier Device for Aerosol Reduction in a Dental Environment: Computational Fluid Dynamics Simulation"

_bioengineering, 2023, doi:10.3390/bioengineering10080947_

Round 1

Reviewer 1 Report

This manuscript reported an air-barrier device to work as protective equipment in the dental clinic. The numerical and experimental results show this device can remove the particles quickly and effectively. The manuscript is well-organized and written. The publication can be recommended after considering the following questions:

  1. In section 2.1, more detail about the mesh is needed, such as structured or unstructured mesh, total meshed in the computational area, and the minimum and maximum. Which molds were used in COMSOL, just CFD mold? Is it a time-dependent or stationary study?

  2. On page 5, please explain why assume the flow is compressible. From my understanding, the air is considered incompressible at low velocity (<100m/s). Does compressed air use in this work? If so, give more detail, such as pressure and temperature.

  3. Please include the references for equation 2.

  4. Check the subscript in the context to make sure it is in the right position.

Reviewer 2 Report

In order to lower contact with contaminants, the author developed a prototype of an air barrier device that forms an air curtain as well as performs suction and evaluated the effect of this newly developed device through a simulation study and experiments. This study demonstrates that air barrier devices in dental environments can play an effective role in reducing contaminating particles.

(1)Why did the author choose k-ε turbulence model for CFD numerical simulation?

(2)The author divides the airflow into 1) potential core,2) transition, and 3) developed zones. Please give the basis of regional differentiation?

(3)Where is the Figuer S1,Figuer S2, Figuer S3 in the article?

(4)Why did the author choose these three injection angles to study?

Reviewer 3 Report

Remove the words like our study and our device.. Use ‘the’.

In several studies, velocity is not constant, Example: Velocity is increasing from 12m/s to 16m/s in Figure 6. Why?

In conclusion, the authors indicate that the study is optimization research. However, there is no specific optimization study throughout the paper while the parameters are investigated. So, clarification is needed.

There is no depth provided on Finite Element Study inside the paper other than Table 3. Details are needed.

Reviewer 4 Report

The paper adresses an interesting topic both from the theoretical (simulation) and experimental point of view.

However, the simulation part is not described and adressed appropriately.

1. The geometry is insufficiently described. There is no point in the paper where the exact dimensions of the air volumes etc. are provided. All plots e.g. 4a are without length dimensions.

2. There are some dimensions provided for the 3D model (figure 1), however I do not get any match to the scatees in figure 3c or figure 4a

3. The boundary conditions are insufficiently described. There are some information provided concerning the inlet flow, but non what so ever to the ventilation hole (outlet). Figure 3b suggests that acutally the outlet is an inlet.

4. There are no information concerning the grid resolution. The wall resultion is provided without units which does not help.

5. Also no study is done to show that the results are grid independent.

6. Reynolds numbers for the overall flow are not provided.

7. Particle Reynolds numbers are not provided.

8. If one has turbulent flow, one ususually has increased dispersion due to the vortices. In that sense it is not clear, that RANS and particle tracking provides appropirate results.

I also have trouble with the experiments.

1. The transparent walls seen in figure 2D do alter the overall flow. There is no good reason why that situation with walls fits to the situation in the simulation Figure 3c and with the final application without walls.

The units in equation do not fit. Left hand (m/s); Right hand (1)

For the results I have no clue what the 280% last line page 6 compare to. What is the reference situation?

Figure 4b. Are these simulation results, plots of the equations 4 and 5. Or are these experimental results?

How do the simulations results fit to the equations and to the experiments?

In Figure 5 and 6 I do not understand what the time is all about. It seems to me that on the long term virtually all designs have an efficiency of about 100%, which means that they are more or less all the same (on the long term). Do explain better what these figures show and how they relate quantitative to the experiments.

The numbers provided on the top of page 11 show that the results cannot be normal distributed (one sigma downwards is already negative). This means you have to elaborate more what you do statistically, e.g. using non parametric tests ...

How do the results in figure 7 relate to the simulation results?

Overall this paper need a massive improvement on the methodological side, after that it might be reconsidered for resubmission.

Reviewer 5 Report

In this study, the prototype device was designed based on the factor of optimal airbarrier design derived through CFD results. The study was conducted into an interesting manner but I didn't found any details or experiments related to the materials. I think that a minimal discussion and analysis must be made on the used materials. Also, I think that the surface properties of the used materials will influence the experimental results.

Also, I didn't know any "material properies" measured (CFU/cm2 ). Please, revise the table 3.

Round 2

Reviewer 1 Report

The publication is recommended.

Author Response

Thank you for your comments.

Reviewer 4 Report

Concerning the answers of the authors:

1. I do understand that one does not want to risk a patent application by disclosing too much details. However, if by purpose the details are lacking that other scientist have a chance to verify or disprove the research presented, I see the basic intention of scientific publishing not given. For that case my recommendation would be to postpone the publishing to the point when the patent application is done.

2. One suggestion is to improve the clearness of presentation to present the 0.5mm slit in figure 1A in a different colour. The contrast in the rendered picture is super low.

3. What is the flow rate/boundary condition for the ventilation hole.

4. and 5. The information provided in the supplement is fine.

6. I am very surprised about how low the Re is. My suggestion is that the authors add the information how the number was calculated. (velocity, density, length, …) If the Re is right the application of the k-epsilon model is not appropriate, because one is in laminar regime.

7. The particle Re sounds reasonable.

8. The particle tracing follows the stream lines and thus is insensitive to possible dispersion increase that is included in the k-epsilon model. Or putting it the other way round: Particle tracing is insensitive to the diffusion constant.

Experiment:

1.       The box can be clearly seen in Figure 2d. For that situation we have a guided flow. Figure 4a shows that the air flow extends beyond the size of the mouth. If you have experimental results that the box is large enough not to interfere these results should be added to the papers supplement.

Concerning the efficiency I am not sure if the view taken by the authors is right. If every system has the very same efficacy on the long term the outcome on the plates should be the same, i.e. as soon as the airjet is switched on the CFUs should be zero. However, the experimental finding is different. 

Figure caption 7 cannot be right:  it says: A/C airjet on; B/D airjet off; A/C shows bio aerosols; B/D show not bio aerosols. Hence, if the airjet/device is off I do not have bio aerosols.

The paper improved, but it still does not comply with scientific standards. Especially the unwillingness to release the information up to the degree to with other scientist can prove or disprove the findings is a reason to vote against publication.

Here a quote:

Karl Popper "A theory which is not refutable by any conceivable event is non-scientific. Irrefutability is not a virtue of a theory (as people often think) but a vice."

Reviewer 5 Report

The manuscript could be published.

Author Response

Thanks for your comments .

Round 3

Reviewer 4 Report

I do highly appreciate that the designs are now shown.

I do appreciate that now some statement is done with respect to the outlet flow. However, the outlet velocities either they make nearly no difference for the final results, then this finding must be included into the paper, or the outlet velocities impact the overall results and then for each the result it must we stated what the outlet velocities are. I presume that the later is the case. In general all information must be give to that detail that other researchers can verify or disprove the findings being published.

In paper the authors write about "buoyancy drag force". While I believe the drag forces to be relevant, I doubt that buoyancy plays any role in this setting. I ask the authors to check their use of scientific terminology. I that regard I am very happy that the authors checked and corrected the Reynolds number.

I do still believe that the particle tracing gives different results to what will be found in the experiment. But, the authors describe what they do, so I hope that readers will spot where the difference lies. This has nothing to do how the drag force is modelled. You would have to add a random force to model the particle dissipation due to the vortices generated by turbulent flow. Particle traces like in figure 5a are very often found in turbulent flow, but are hardly seen in the particle tracing done are described here in the paper. The reason is that the time dependent vortices typical flow turbulent flow are missing in this simulation.

The explanation that the authors checked that the walls in the experiment do not alter the flow conditions should be added to the paper.

One point that gets clearer to me is, that most likely "Removal Efficiency" is a misnomer. As it is stated in Figure 5a (actually such formulae belong into the text and not into a figure) it would make sense. However, usually simulation programmes remove particles once the get to near to the wall or reach nearly zero velocity. Hence Nob is most likely nearly zero, which leads to the that the efficiency ends up being 1. The other point is that initially the efficiency is an ill-defined property, because at the start no particles are removed thus Nvent=0 and not particles stick anywhere thus Nob=0. Therefore Nvent+Nob=0 leaving a zero in the denominator.

I presume that the denominator used is the number of particles injected. Since no particles are removed at the start this gets a efficiency of zero. My suggestion is that the authors redefine their efficiency to Nvent/Ninjected and then to really count the particles at the vent outlet. This will require some programming but should be feasible. This should lead to different numbers. I also suggest switching from "removal efficiency (%)" to "vented particles (%)". I assume that this rate will depend on the outlet flow (venting flow). If the venting flow is zero the new removal efficiency and then vented particles has to be zero, because not particles leave via the vent (zero venting flow). Figure 6S and 7S is suggesting this already.

Concerning figure 5S my suggestion is to try to visualize with stream lines. In that case it gets clear where the flow is going. The super small arrows are virtually impossible to read without any magnification on the screen. One could also add the points to these graphs where the particles are injected.

Is it right that the simulation are done under the assumption that no air is exhaled or inhaled via the mouth?

My suggestion is that figures like 5S are added to visualize also the effect of the different angles.

The paper is continuously improving, but it is still not really great.

Round 4

Reviewer 4 Report

The authors improved the paper quite some bit again. Especially in combination with the further extended supplementary material I do find not major points that should be improved. My questions and remarks I have found to be addressed well.